# Toward Multimodal Image-to-Image Translation

**Jun-Yan Zhu**
UC Berkeley

**Richard Zhang**
UC Berkeley

**Deepak Pathak**
UC Berkeley

**Trevor Darrell**
UC Berkeley

**Alexei A. Efros**
UC Berkeley

**Oliver Wang**
Adobe Research

**Eli Shechtman**
Adobe Research

## Abstract

Many image-to-image translation problems are ambiguous, as a single input image may correspond to multiple possible outputs. In this work, we aim to model a *distribution* of possible outputs in a conditional generative modeling setting. The ambiguity of the mapping is distilled in a low-dimensional latent vector, which can be randomly sampled at test time. A generator learns to map the given input, combined with this latent code, to the output. We explicitly encourage the connection between output and the latent code to be invertible. This helps prevent a many-to-one mapping from the latent code to the output during training, also known as the problem of mode collapse, and produces more diverse results. We explore several variants of this approach by employing different training objectives, network architectures, and methods of injecting the latent code. Our proposed method encourages bijective consistency between the latent encoding and output modes. We present a systematic comparison of our method and other variants on both perceptual realism and diversity.

## 1 Introduction

Deep learning techniques have made rapid progress in conditional image generation. For example, networks have been used to inpaint missing image regions [20, 34, 47], add color to grayscale images [19, 20, 27, 50], and generate photorealistic images from sketches [20, 40]. However, most techniques in this space have focused on generating a *single* result. In this work, we model a *distribution* of potential results, as many of these problems may be multimodal in nature. For example, as seen in Figure 1, an image captured at night may look very different in the day, depending on cloud patterns and lighting conditions. We pursue two main goals: producing results which are (1) perceptually realistic and (2) diverse, all while remaining faithful to the input.

Mapping from a high-dimensional input to a high-dimensional output distribution is challenging. A common approach to representing multimodality is learning a low-dimensional latent code, which should represent aspects of the possible outputs not contained in the input image. At inference time, a deterministic generator uses the input image, along with stochastically sampled latent codes, to produce randomly sampled outputs. A common problem in existing methods is *mode collapse* [14], where only a small number of real samples get represented in the output. We systematically study a family of solutions to this problem.

We start with the `pix2pix` framework [20], which has previously been shown to produce high-quality results for various image-to-image translation tasks. The method trains a generator network, conditioned on the input image, with two losses: (1) a regression loss to produce similar output to the known paired ground truth image and (2) a learned discriminator loss to encourage realism. The authors note that trivially appending a randomly drawn latent code did not produce diverse results. Instead, we propose encouraging a bijection between the output and latent space. We not

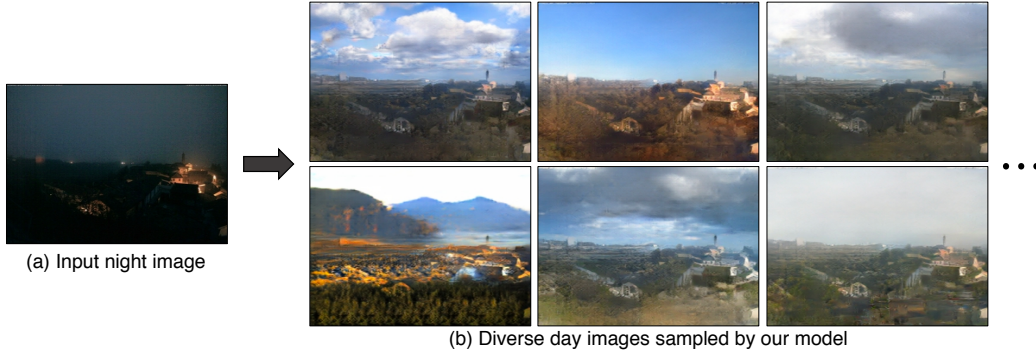

(a) Input night image

(b) Diverse day images sampled by our model

Figure 1: Multimodal image-to-image translation using our proposed method: given an input image from one domain (night image of a scene), we aim to model a *distribution* of potential outputs in the target domain (corresponding day images), producing both realistic and diverse results.

only perform the direct task of mapping the latent code (along with the input) to the output but also jointly learn an encoder from the output back to the latent space. This discourages two different latent codes from generating the same output (non-injective mapping). During training, the learned encoder attempts to pass enough information to the generator to resolve any ambiguities regarding the output mode. For example, when generating a day image from a night image, the latent vector may encode information about the sky color, lighting effects on the ground, and cloud patterns. Composing the encoder and generator sequentially should result in the same image being recovered. The opposite should produce the same latent code.

In this work, we instantiate this idea by exploring several objective functions, inspired by literature in unconditional generative modeling:

- `cVAE-GAN` (**Conditional Variational Autoencoder GAN**): One approach is first encoding the ground truth image into the latent space, giving the generator a noisy "peek" into the desired output. Using this, along with the input image, the generator should be able to reconstruct the specific output image. To ensure that random sampling can be used during inference time, the latent distribution is regularized using KL-divergence to be close to a standard normal distribution. This approach has been popularized in the unconditional setting by VAEs [23] and VAE-GANs [26].

- `cLR-GAN` (**Conditional Latent Regressor GAN**): Another approach is to first provide a randomly drawn latent vector to the generator. In this case, the produced output may not necessarily look like the ground truth image, but it should look realistic. An encoder then attempts to recover the latent vector from the output image. This method could be seen as a conditional formulation of the "latent regressor" model [8, 10] and also related to InfoGAN [4].

- `BicycleGAN`: Finally, we combine both these approaches to enforce the connection between latent encoding and output in both directions *jointly* and achieve improved performance. We show that our method can produce both diverse and visually appealing results across a wide range of image-to-image translation problems, significantly more diverse than other baselines, including naively adding noise in the `pix2pix` framework. In addition to the loss function, we study the performance with respect to several encoder networks, as well as different ways of injecting the latent code into the generator network.

We perform a systematic evaluation of these variants by using humans to judge photorealism and a perceptual distance metric [52] to assess output diversity. Code and data are available at `https://github.com/junyanz/BicycleGAN`.

## 2  Related Work

**Generative modeling**   Parametric modeling of the natural image distribution is a challenging problem. Classically, this problem has been tackled using restricted Boltzmann machines [41] and autoencoders [18, 43]. Variational autoencoders [23] provide an effective approach for modeling stochasticity within the network by reparametrization of a latent distribution at training time. A different approach is autoregressive models [11, 32, 33], which are effective at modeling natural

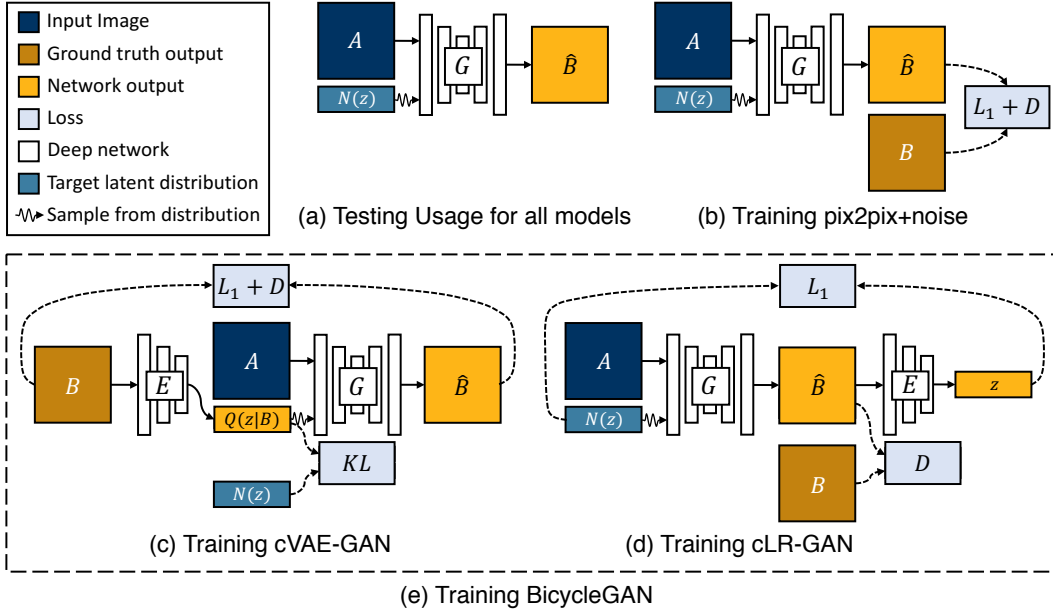

Figure 2: **Overview**: (a) Test time usage of all the methods. To produce a sample output, a latent code **z** is first randomly sampled from a known distribution (e.g., a standard normal distribution). A generator $G$ maps an input image **A** (blue) and the latent sample **z** to produce a output sample $\hat{\mathbf{B}}$ (yellow). (b) pix2pix+noise [20] baseline, with an additional ground truth image **B** (brown) that corresponds to **A**. (c) cVAE-GAN (and cAE-GAN) starts from a ground truth target image **B** and encode it into the latent space. The generator then attempts to map the input image **A** along with a sampled **z** back into the original image **B**. (d) cLR-GAN randomly samples a latent code from a known distribution, uses it to map **A** into the output $\hat{\mathbf{B}}$, and then tries to reconstruct the latent code from the output. (e) Our hybrid BicycleGAN method combines constraints in both directions.

image statistics but are slow at inference time due to their sequential predictive nature. Generative adversarial networks [15] overcome this issue by mapping random values from an easy-to-sample distribution (e.g., a low-dimensional Gaussian) to output images in a single feedforward pass of a network. During training, the samples are judged using a discriminator network, which distinguishes between samples from the target distribution and the generator network. GANs have recently been very successful [1, 4, 6, 8, 10, 35, 36, 49, 53, 54]. Our method builds on the conditional version of VAE [23] and InfoGAN [4] or latent regressor [8, 10] models by jointly optimizing their objectives. We revisit this connection in Section 3.4.

**Conditional image generation**     All of the methods defined above can be easily conditioned. While conditional VAEs [42] and autoregressive models [32, 33] have shown promise [16, 44, 46], image-to-image conditional GANs have lead to a substantial boost in the quality of the results. However, the quality has been attained at the expense of multimodality, as the generator learns to largely ignore the random noise vector when conditioned on a relevant context [20, 34, 40, 45, 47, 55]. In fact, it has even been shown that ignoring the noise leads to more stable training [20, 29, 34].

**Explicitly-encoded multimodality**     One way to express multiple modes is to explicitly encode them, and provide them as an additional input in addition to the input image. For example, color and shape scribbles and other interfaces were used as conditioning in iGAN [54], pix2pix [20], Scribbler [40] and interactive colorization [51]. An effective option explored by concurrent work [2, 3, 13] is to use a mixture of models. Though able to produce multiple discrete answers, these methods are unable to produce continuous changes. While there has been some degree of success for generating multimodal outputs in unconditional and text-conditional setups [7, 15, 26, 31, 36], conditional image-to-image generation is still far from achieving the same results, unless explicitly encoded as discussed above. In this work, we learn conditional image generation models for modeling multiple modes of output by enforcing tight connections between the latent and image spaces.

# 3 Multimodal Image-to-Image Translation

Our goal is to learn a multi-modal mapping between two image domains, for example, edges and photographs, or night and day images, etc. Consider the input domain $\mathcal{A} \subset \mathbb{R}^{H \times W \times 3}$, which is to be mapped to an output domain $\mathcal{B} \subset \mathbb{R}^{H \times W \times 3}$. During training, we are given a dataset of paired instances from these domains, $\{(\mathbf{A} \in \mathcal{A}, \mathbf{B} \in \mathcal{B})\}$, which is representative of a joint distribution $p(\mathbf{A}, \mathbf{B})$. It is important to note that there could be multiple plausible paired instances $\mathbf{B}$ that would correspond to an input instance $\mathbf{A}$, but the training dataset usually contains only one such pair. However, given a new instance $\mathbf{A}$ during test time, our model should be able to generate a diverse set of output $\widehat{\mathbf{B}}$'s, corresponding to different modes in the distribution $p(\mathbf{B}|\mathbf{A})$.

While conditional GANs have achieved success in image-to-image translation tasks [20, 34, 40, 45, 47, 55], they are primarily limited to generating a deterministic output $\widehat{\mathbf{B}}$ given the input image $\mathbf{A}$. On the other hand, we would like to learn the mapping that could sample the output $\widehat{\mathbf{B}}$ from true conditional distribution given $\mathbf{A}$, and produce results which are both diverse and realistic. To do so, we learn a low-dimensional latent space $\mathbf{z} \in \mathbb{R}^Z$, which encapsulates the ambiguous aspects of the output mode which are not present in the input image. For example, a sketch of a shoe could map to a variety of colors and textures, which could get compressed in this latent code. We then learn a deterministic mapping $G : (\mathbf{A}, \mathbf{z}) \to \mathbf{B}$ to the output. To enable stochastic sampling, we desire the latent code vector $\mathbf{z}$ to be drawn from some prior distribution $p(\mathbf{z})$; we use a standard Gaussian distribution $\mathcal{N}(0, I)$ in this work.

We first discuss a simple extension of existing methods and discuss its strengths and weakness, motivating the development of our proposed approach in the subsequent subsections.

## 3.1 Baseline: `pix2pix+noise` $(\mathbf{z} \to \widehat{\mathbf{B}})$

The recently proposed `pix2pix` model [20] has shown high quality results in the image-to-image translation setting. It uses conditional adversarial networks [15, 30] to help produce perceptually realistic results. GANs train a generator $G$ and discriminator $D$ by formulating their objective as an adversarial game. The discriminator attempts to differentiate between real images from the dataset and fake samples produced by the generator. Randomly drawn noise $\mathbf{z}$ is added to attempt to induce stochasticity. We illustrate the formulation in Figure 2(b) and describe it below.

$$\mathcal{L}_{\text{GAN}}(G, D) = \mathbb{E}_{\mathbf{A}, \mathbf{B} \sim p(\mathbf{A}, \mathbf{B})}[\log(D(\mathbf{A}, \mathbf{B}))] + \mathbb{E}_{\mathbf{A} \sim p(\mathbf{A}), \mathbf{z} \sim p(\mathbf{z})}[\log(1 - D(\mathbf{A}, G(\mathbf{A}, \mathbf{z})))] \quad (1)$$

To encourage the output of the generator to match the input as well as stabilize the training, we use an $\ell_1$ loss between the output and the ground truth image.

$$\mathcal{L}_1^{\text{image}}(G) = \mathbb{E}_{\mathbf{A}, \mathbf{B} \sim p(\mathbf{A}, \mathbf{B}), \mathbf{z} \sim p(\mathbf{z})}||\mathbf{B} - G(\mathbf{A}, \mathbf{z})||_1 \quad (2)$$

The final loss function uses the GAN and $\ell_1$ terms, balanced by $\lambda$.

$$G^* = \arg\min_G \max_D \quad \mathcal{L}_{\text{GAN}}(G, D) + \lambda \mathcal{L}_1^{\text{image}}(G) \quad (3)$$

In this scenario, there is little incentive for the generator to make use of the noise vector which encodes random information. Isola et al. [20] note that the noise was ignored by the generator in preliminary experiments and was removed from the final experiments. This was consistent with observations made in the conditional settings by [29, 34], as well as the mode collapse phenomenon observed in unconditional cases [14, 39]. In this paper, we explore different ways to explicitly enforce the latent coding to capture relevant information.

## 3.2 Conditional Variational Autoencoder GAN: `cVAE-GAN` $(\mathbf{B} \to \mathbf{z} \to \widehat{\mathbf{B}})$

One way to force the latent code $\mathbf{z}$ to be "useful" is to directly map the ground truth $\mathbf{B}$ to it using an encoding function $E$. The generator $G$ then uses both the latent code and the input image $\mathbf{A}$ to synthesize the desired output $\widehat{\mathbf{B}}$. The overall model can be easily understood as the reconstruction of $\mathbf{B}$, with latent encoding $\mathbf{z}$ concatenated with the paired $\mathbf{A}$ in the middle – similar to an autoencoder [18]. This interpretation is better shown in Figure 2(c).

This approach has been successfully investigated in Variational Autoencoder [23] in the unconditional scenario without the adversarial objective. Extending it to conditional scenario, the distribution $Q(\mathbf{z}|\mathbf{B})$ of latent code $\mathbf{z}$ using the encoder $E$ with a Gaussian assumption, $Q(\mathbf{z}|\mathbf{B}) \triangleq E(\mathbf{B})$. To reflect this, Equation 1 is modified to sampling $\mathbf{z} \sim E(\mathbf{B})$ using the re-parameterization trick, allowing direct back-propagation [23].

$$\mathcal{L}_{\text{GAN}}^{\text{VAE}} = \mathbb{E}_{\mathbf{A},\mathbf{B}\sim p(\mathbf{A},\mathbf{B})}[\log(D(\mathbf{A},\mathbf{B}))] + \mathbb{E}_{\mathbf{A},\mathbf{B}\sim p(\mathbf{A},\mathbf{B}),\mathbf{z}\sim E(\mathbf{B})}[\log(1 - D(\mathbf{A}, G(\mathbf{A},\mathbf{z})))] \quad (4)$$

We make the corresponding change in the $\ell_1$ loss term in Equation 2 as well to obtain $\mathcal{L}_1^{\text{VAE}}(G) = \mathbb{E}_{\mathbf{A},\mathbf{B}\sim p(\mathbf{A},\mathbf{B}),\mathbf{z}\sim E(\mathbf{B})}||\mathbf{B} - G(\mathbf{A},\mathbf{z})||_1$. Further, the latent distribution encoded by $E(B)$ is encouraged to be close to a random Gaussian to enable sampling at inference time, when $\mathbf{B}$ is not known.

$$\mathcal{L}_{\text{KL}}(E) = \mathbb{E}_{\mathbf{B}\sim p(\mathbf{B})}[\mathcal{D}_{\text{KL}}(E(\mathbf{B})|| \mathcal{N}(0,I))], \quad (5)$$

where $\mathcal{D}_{\text{KL}}(p||q) = -\int p(z) \log \frac{p(z)}{q(z)} dz$. This forms our `cVAE-GAN` objective, a conditional version of the VAE-GAN [26] as

$$G^*, E^* = \arg\min_{G,E} \max_{D} \quad \mathcal{L}_{\text{GAN}}^{\text{VAE}}(G, D, E) + \lambda\mathcal{L}_1^{\text{VAE}}(G, E) + \lambda_{\text{KL}}\mathcal{L}_{\text{KL}}(E). \quad (6)$$

As a baseline, we also consider the deterministic version of this approach, i.e., dropping KL-divergence and encoding $\mathbf{z} = E(\mathbf{B})$. We call it `cAE-GAN` and show a comparison in the experiments. There is no guarantee in `cAE-GAN` on the distribution of the latent space $\mathbf{z}$, which makes the test-time sampling of $\mathbf{z}$ difficult.

### 3.3  Conditional Latent Regressor GAN: `cLR-GAN` ($\mathbf{z} \rightarrow \widehat{\mathbf{B}} \rightarrow \widehat{\mathbf{z}}$)

We explore another method of enforcing the generator network to utilize the latent code embedding $\mathbf{z}$, while staying close to the actual test time distribution $p(\mathbf{z})$, but from the latent code's perspective. As shown in Figure 2(d), we start from a randomly drawn latent code $\mathbf{z}$ and attempt to recover it with $\widehat{\mathbf{z}} = E(G(\mathbf{A},\mathbf{z}))$. Note that the encoder $E$ here is producing a point estimate for $\widehat{\mathbf{z}}$, whereas the encoder in the previous section was predicting a Gaussian distribution.

$$\mathcal{L}_1^{\text{latent}}(G, E) = \mathbb{E}_{\mathbf{A}\sim p(\mathbf{A}),\mathbf{z}\sim p(\mathbf{z})}||\mathbf{z} - E(G(\mathbf{A},\mathbf{z}))||_1 \quad (7)$$

We also include the discriminator loss $L_{\text{GAN}}(G, D)$ (Equation 1) on $\widehat{\mathbf{B}}$ to encourage the network to generate realistic results, and the full loss can be written as:

$$G^*, E^* = \arg\min_{G,E} \max_{D} \quad \mathcal{L}_{\text{GAN}}(G, D) + \lambda_{\text{latent}}\mathcal{L}_1^{\text{latent}}(G, E) \quad (8)$$

The $\ell_1$ loss for the ground truth image $\mathbf{B}$ is not used. Since the noise vector is randomly drawn, the predicted $\widehat{\mathbf{B}}$ does not necessarily need to be close to the ground truth but does need to be realistic. The above objective bears similarity to the "latent regressor" model [4, 8, 10], where the generated sample $\widehat{\mathbf{B}}$ is encoded to generate a latent vector.

### 3.4  Our Hybrid Model: `BicycleGAN`

We combine the `cVAE-GAN` and `cLR-GAN` objectives in a hybrid model. For `cVAE-GAN`, the encoding is learned from real data, but a random latent code may not yield realistic images at test time – the KL loss may not be well optimized. Perhaps more importantly, the adversarial classifier $D$ does not have a chance to see results sampled from the prior during training. In `cLR-GAN`, the latent space is easily sampled from a simple distribution, but the generator is trained without the benefit of seeing ground truth input-output pairs. We propose to train with constraints in both directions, aiming to take advantage of both cycles ($\mathbf{B} \rightarrow \mathbf{z} \rightarrow \widehat{\mathbf{B}}$ and $\mathbf{z} \rightarrow \widehat{\mathbf{B}} \rightarrow \widehat{\mathbf{z}}$), hence the name `BicycleGAN`.

$$\begin{aligned} G^*, E^* = \arg\min_{G,E} \max_{D} \quad & \mathcal{L}_{\text{GAN}}^{\text{VAE}}(G, D, E) + \lambda\mathcal{L}_1^{\text{VAE}}(G, E) \\ & + \mathcal{L}_{\text{GAN}}(G, D) + \lambda_{\text{latent}}\mathcal{L}_1^{\text{latent}}(G, E) + \lambda_{\text{KL}}\mathcal{L}_{\text{KL}}(E), \end{aligned} \quad (9)$$

where the hyper-parameters $\lambda$, $\lambda_{\text{latent}}$, and $\lambda_{\text{KL}}$ control the relative importance of each term.

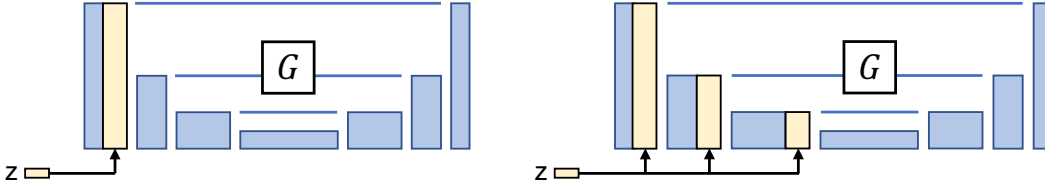

Figure 3: **Alternatives for injecting z into generator.** Latent code **z** is injected by spatial replication and concatenation into the generator network. We tried two alternatives, **(left)** injecting into the input layer and **(right)** every intermediate layer in the encoder.

In the unconditional GAN setting, Larsen et al. [26] observe that using samples from both the prior $\mathcal{N}(0, I)$ and encoded $E(\mathbf{B})$ distributions further improves results. Hence, we also report one variant which is the full objective shown above (Equation 9), but without the reconstruction loss on the latent space $\mathcal{L}_1^{\text{latent}}$. We call it cVAE-GAN++, as it is based on cVAE-GAN with an additional loss $\mathcal{L}_{\text{GAN}}(G, D)$, which allows the discriminator to see randomly drawn samples from the prior.

## 4 Implementation Details

The code and additional results are publicly available at `https://github.com/junyanz/BicycleGAN`. Please refer to our website for more details about the datasets, architectures, and training procedures.

**Network architecture**    For generator $G$, we use the U-Net [37], which contains an encoder-decoder architecture, with symmetric skip connections. The architecture has been shown to produce strong results in the unimodal image prediction setting when there is a spatial correspondence between input and output pairs. For discriminator $D$, we use two PatchGAN discriminators [20] at different scales, which aims to predict real vs. fake for $70 \times 70$ and $140 \times 140$ overlapping image patches. For the encoder $E$, we experiment with two networks: (1) $E_{\text{CNN}}$: CNN with a few convolutional and downsampling layers and (2) $E_{\text{ResNet}}$: a classifier with several residual blocks [17].

**Training details**    We build our model on the Least Squares GANs (LSGANs) variant [28], which uses a least-squares objective instead of a cross entropy loss. LSGANs produce high-quality results with stable training. We also find that not conditioning the discriminator $D$ on input $\mathbf{A}$ leads to better results (also discussed in [34]), and hence choose to do the same for all methods. We set the parameters $\lambda_{\text{image}} = 10$, $\lambda_{\text{latent}} = 0.5$ and $\lambda_{\text{KL}} = 0.01$ in all our experiments. We tie the weights for the generators and encoders in the cVAE-GAN and cLR-GAN models. For the encoder, only the predicted mean is used in cLR-GAN. We observe that using two separate discriminators yields slightly better visual results compared to sharing weights. We only update $G$ for the $\ell_1$ loss $\mathcal{L}_1^{\text{latent}}(G, E)$ on the latent code (Equation 7), while keeping $E$ fixed. We found optimizing $G$ and $E$ simultaneously for the loss would encourage $G$ and $E$ to hide the information of the latent code without learning meaningful modes. We train our networks from scratch using Adam [22] with a batch size of 1 and with a learning rate of 0.0002. We choose latent dimension $|\mathbf{z}| = 8$ across all the datasets.

**Injecting the latent code z to generator**.    We explore two ways of propagating the latent code **z** to the output, as shown in Figure 3: (1) `add_to_input`: we spatially replicate a $Z$-dimensional latent code **z** to an $H \times W \times Z$ tensor and concatenate it with the $H \times W \times 3$ input image and (2) `add_to_all`: we add **z** to each intermediate layer of the network $G$, after spatial replication to the appropriate sizes.

## 5 Experiments

**Datasets**    We test our method on several image-to-image translation problems from prior work, including edges $\rightarrow$ photos [48, 54], Google maps $\rightarrow$ satellite [20], labels $\rightarrow$ images [5], and outdoor night $\rightarrow$ day images [25]. These problems are all one-to-many mappings. We train all the models on $256 \times 256$ images.

**Methods**    We evaluate the following models described in Section 3: `pix2pix+noise`, `cAE-GAN`, `cVAE-GAN`, `cVAE-GAN++`, `cLR-GAN`, and our hybrid model `BicycleGAN`.

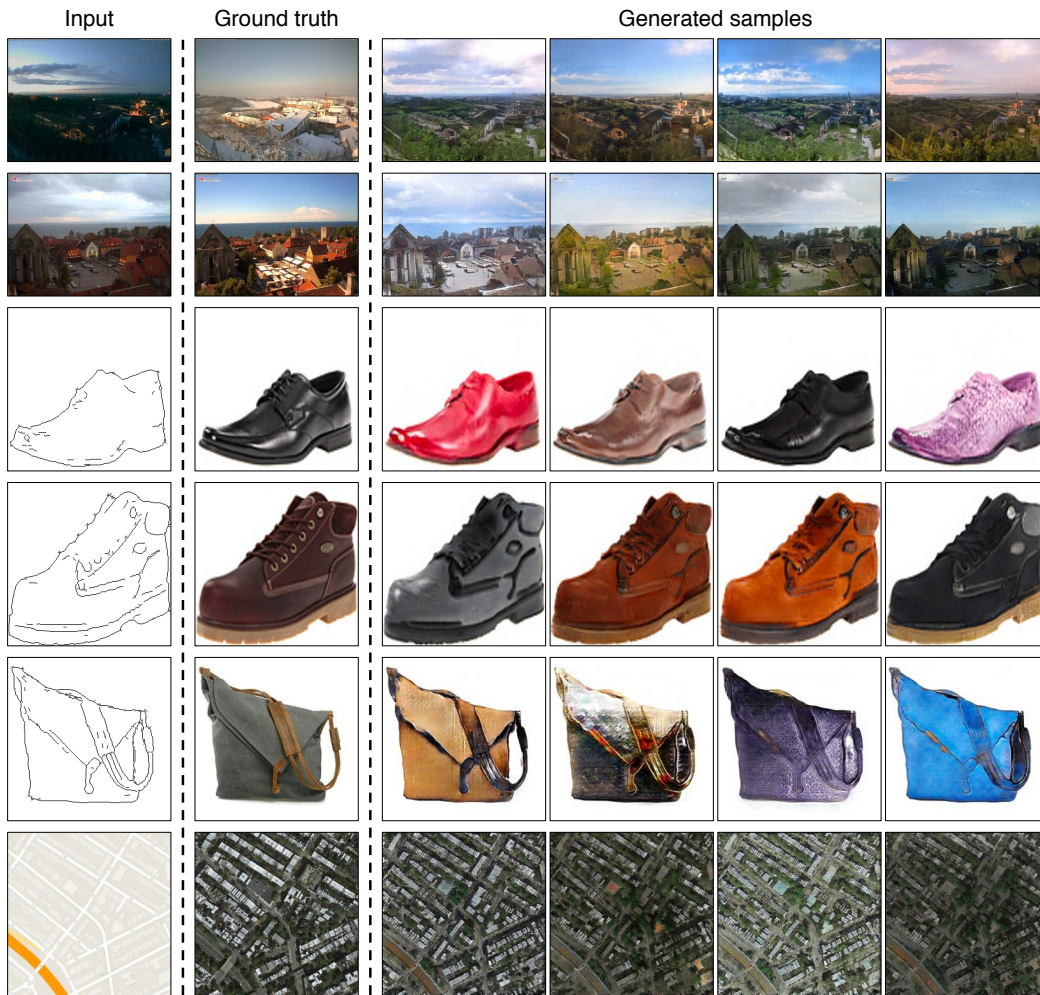

Figure 4: **Example Results** We show example results of our hybrid model `BicycleGAN`. The left column shows the input. The second shows the ground truth output. The final four columns show randomly generated samples. We show results of our method on night→day, edges→shoes, edges→handbags, and maps→satellites. Models and additional examples are available at `https://junyanz.github.io/BicycleGAN`.

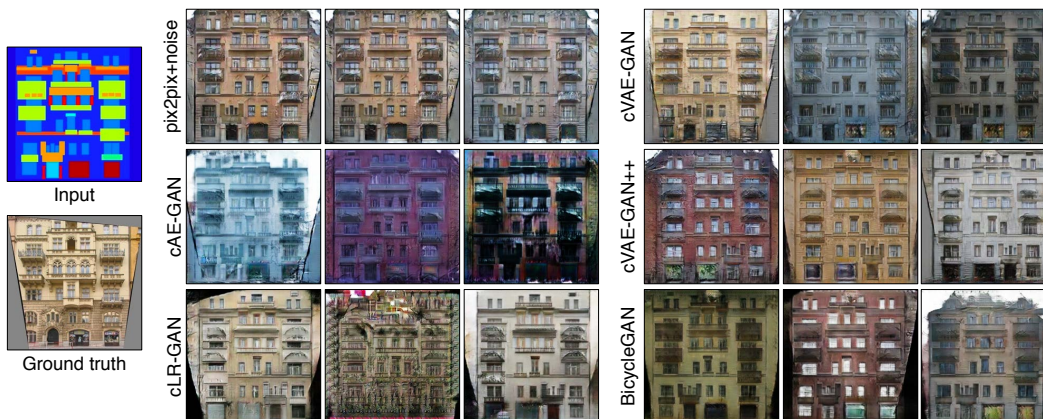

Figure 5: **Qualitative method comparison** We compare results on the labels → facades dataset across different methods. The `BicycleGAN` method produces results which are both realistic and diverse.

| | Realism | Diversity |
|---|---|---|
| | AMT Fooling | LPIPS |
| **Method** | Rate [%] | Distance |
| Random real images | 50.0% | .265±.007 |
| `pix2pix+noise` [20] | 27.93±2.40 % | .013±.000 |
| `cAE-GAN` | 13.64±1.80 % | .200±.002 |
| `cVAE-GAN` | 24.93±2.27 % | .095±.001 |
| `cVAE-GAN++` | 29.19±2.43 % | .099±.002 |
| `cLR-GAN` | 29.23±2.48 % | [a].089±.002 |
| `BicycleGAN` | 34.33±2.69 % | .111±.002 |

[a]We found that `cLR-GAN` resulted in severe mode collapse, resulting in ~ 15% of the images producing the same result. Those images were omitted from this calculation.

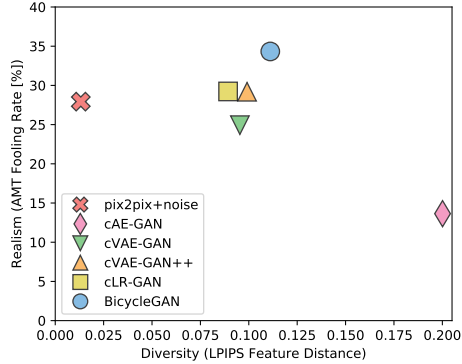

Figure 6: **Realism vs Diversity**. We measure diversity using average LPIPS distance [52], and realism using a real vs. fake Amazon Mechanical Turk test on the Google maps → satellites task. The `pix2pix+noise` baseline produces little diversity. Using only `cAE-GAN` method produces large artifacts during sampling. The hybrid `BicycleGAN` method, which combines `cVAE-GAN` and `cLR-GAN`, produces results which have higher realism while maintaining diversity.

## 5.1 Qualitative Evaluation

We show qualitative comparison results on Figure 5. We observe that `pix2pix+noise` typically produces a single realistic output, but does not produce any meaningful variation. `cAE-GAN` adds variation to the output, but typically at a large cost to result quality. An example on facades is shown on Figure 4.

We observe more variation in the `cVAE-GAN`, as the latent space is encouraged to encode information about ground truth outputs. However, the space is not densely populated, so drawing random samples may cause artifacts in the output. The `cLR-GAN` shows less variation in the output, and sometimes suffers from mode collapse. When combining these methods, however, in the hybrid method `BicycleGAN`, we observe results which are both diverse and realistic. Please see our website for a full set of results.

## 5.2 Quantitative Evaluation

We perform a quantitative analysis of the diversity, realism, and latent space distribution on our six variants and baselines. We quantitatively test the Google maps → satellites dataset.

**Diversity** We compute the average distance of random samples in deep feature space. Pretrained networks have been used as a "perceptual loss" in image generation applications [9, 12, 21], as well as a held-out "validation" score in generative modeling, for example, assessing the semantic quality and diversity of a generative model [39] or the semantic accuracy of a grayscale colorization [50].

In Figure 6, we show the diversity-score using the LPIPS metric proposed by [52][1]. For each method, we compute the average distance between 1900 pairs of randomly generated output $\widehat{\mathbf{B}}$ images (sampled from 100 input $\mathbf{A}$ images). Random pairs of ground truth real images in the $\mathbf{B} \in \mathcal{B}$ domain produce an average variation of .265. As we are measuring samples $\widehat{\mathbf{B}}$ which correspond to a specific input $\mathbf{A}$, a system which stays faithful to the input should definitely not exceed this score.

The `pix2pix` system [20] produces a single point estimate. Adding noise to the system `pix2pix+noise` produces a small diversity score, confirming the finding in [20] that adding noise does not produce large variation. Using the `cAE-GAN` model to encode a ground truth image $\mathbf{B}$ into a latent code $\mathbf{z}$ does increase the variation. The `cVAE-GAN`, `cVAE-GAN++`, and `BicycleGAN` models all place explicit constraints on the latent space, and the `cLR-GAN` model places an implicit constraint through sampling. These four methods all produce similar diversity scores. We note that high diversity scores may also indicate that unnatural images are being generated, causing meaningless variations. Next, we investigate the visual realism of our samples.

**Perceptual Realism** To judge the visual realism of our results, we use human judgments, as proposed in [50] and later used in [20, 55]. The test sequentially presents a real and generated image to a human

| Encoder | $E_{\tt ResNet}$ | $E_{\tt ResNet}$ | $E_{\tt CNN}$ | $E_{\tt CNN}$ |
|---|---|---|---|---|
| Injecting $\mathbf{z}$ | add_to_all | add_to_input | add_to_all | add_to_input |
| label→photo | $0.292 \pm 0.058$ | $0.292 \pm 0.054$ | $0.326 \pm 0.066$ | $0.339 \pm 0.069$ |
| map $\rightarrow$ satellite | $0.268 \pm 0.070$ | $0.266 \pm 0.068$ | $0.287 \pm 0.067$ | $0.272 \pm 0.069$ |

Table 1: The encoding performance with respect to the different encoder architectures and methods of injecting $\mathbf{z}$. Here we report the reconstruction loss $||\mathbf{B} - G(\mathbf{A}, E(B))||_1$

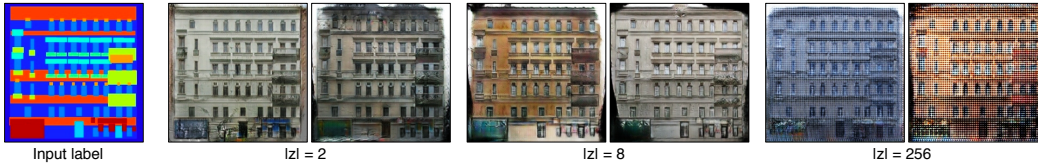

Figure 7: Different label $\rightarrow$ facades results trained with varying length of the latent code $|\mathbf{z}| \in \{2, 8, 256\}$.

for 1 second each, in a random order, asks them to identify the fake, and measures the "fooling" rate. Figure 6(left) shows the realism across methods. The `pix2pix+noise` model achieves high realism score, but without large diversity, as discussed in the previous section. The `cAE-GAN` helps produce diversity, but this comes at a large cost to the visual realism. Because the distribution of the learned latent space is unclear, random samples may be from unpopulated regions of the space. Adding the KL-divergence loss in the latent space, used in the `cVAE-GAN` model, recovers the visual realism. Furthermore, as expected, checking randomly drawn $\mathbf{z}$ vectors in the `cVAE-GAN++` model slightly increases realism. The `cLR-GAN`, which draws $\mathbf{z}$ vectors from the predefined distribution randomly, produces similar realism and diversity scores. However, the `cLR-GAN` model resulted in large mode collapse - approximately $15\%$ of the outputs produced the same result, independent of the input image. The full hybrid `BicycleGAN` gets the best of both worlds, as it does not suffer from mode collapse and also has the highest realism score by a significant margin.

**Encoder architecture** In `pix2pix`, Isola et al. [20] conduct extensive ablation studies on discriminators and generators. Here we focus on the performance of two encoder architectures, $E_{\tt CNN}$ and $E_{\tt ResNet}$, for our applications on the maps and facades datasets. We find that $E_{\tt ResNet}$ better encodes the output image, regarding the image reconstruction loss $||\mathbf{B} - G(\mathbf{A}, E(B))||_1$ on validation datasets as shown in Table 1. We use $E_{\tt ResNet}$ in our final model.

**Methods of injecting latent code** We evaluate two ways of injecting latent code $\mathbf{z}$: add_to_input and add_to_all (Section 4), regarding the same reconstruction loss $||\mathbf{B} - G(\mathbf{A}, E(B))||_1$. Table 1 shows that two methods give similar performance. This indicates that the U_Net [37] can already propagate the information well to the output without the additional skip connections from $\mathbf{z}$. We use add_to_all method to inject noise in our final model.

**Latent code length** We study the `BicycleGAN` model results with respect to the varying number of dimensions of latent codes $\{2, 8, 256\}$ in Figure 7. A very low-dimensional latent code may limit the amount of diversity that can be expressed. On the contrary, a very high-dimensional latent code can potentially encode more information about an output image, at the cost of making sampling difficult. The optimal length of $\mathbf{z}$ largely depends on individual datasets and applications, and how much ambiguity there is in the output.

## 6 Conclusions

In conclusion, we have evaluated a few methods for combating the problem of mode collapse in the conditional image generation setting. We find that by combining multiple objectives for encouraging a bijective mapping between the latent and output spaces, we obtain results which are more realistic and diverse. We see many interesting avenues of future work, including directly enforcing a distribution in the latent space that encodes semantically meaningful attributes to allow for image-to-image transformations with user controllable parameters.

**Acknowledgments** We thank Phillip Isola and Tinghui Zhou for helpful discussions. This work was supported in part by Adobe Inc., DARPA, AFRL, DoD MURI award N000141110688, NSF awards IIS-1633310, IIS-1427425, IIS-1212798, the Berkeley Artificial Intelligence Research (BAIR) Lab, and hardware donations from NVIDIA. JYZ is supported by Facebook Graduate Fellowship, RZ by Adobe Research Fellowship, and DP by NVIDIA Graduate Fellowship.

## Footnotes

[1]Learned Perceptual Image Patch Similarity (LPIPS) metric computes distance in AlexNet [24] feature space (`conv1-5`, pretrained on Imagenet [38]), with linear weights to better match human perceptual judgments.

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
