[Reviews · NeurIPS 2017]

Reviewer 1



This paper proposes a novel way to generate diverse samples in a conditional generative modeling setting by enforcing cycle consistency on a latent space. The task is translating an image of domain A to another domain while making the results perceptually realistic and diverse. The image cycle consistency helps the model to generate more realistic images while the latent cycle consistency enforces the latent space and the image space to have one-to-one mapping, which means the latent code can effectively embed diverse color and texture information. [Paper strengths] - This paper alleviates the mode collapse problem occurred in the prior work and generates diverse samples with latent codes. - The proposed method is novel and the results are convincing. - Supporting experiments shown in Fig. 3 and Fig. 4 explain well how Bicycle consistency improves diversity of results. [Paper weaknesses] - The paper contains inaccurate formulas in equation (3), (5), (7) and (8). Since GAN losses in (1) and (4) become small if the discriminator accurately discriminates real and fake images, “arg min(G) max(D)” should be “arg max(G) min(D)”. Also, the equation (5), (7), (8) miss an encoder “E” in “min”. - It would be better to show that the changes in diversity w.r.t the increasing dimensions of latent variables. [Clarity] The paper is clearly written and the Fig 1 is helpful for understanding the overall architecture. [Reproducibility] Dataset used for evaluation is publicly available. Even though the paper omits some details about the experiments and contains incorrect equations, the method can be reproduced since the proposed model shares the architecture with prior work. However, it would be better to elaborate missing details (e.g., hyperparameters, training details). [Overall] Explain your rating by discussing the strengths and weaknesses of the submission, contributions, and the potential impact of the paper. Include suggestions for improvement and publication alternatives, if appropriate. Be thorough. Be Fair. Be courteous. Your evaluation will be forwarded to the authors during the rebuttal period. The main contribution of this paper is adding a latent code to generate diverse samples in an image-to-image translation task and using latent cycle-consistency to prevent the model to ignore the latent code. Although there are some incorrect equations, but they are minor. The method is well explained and justified.

Reviewer 2



This paper proposes BicycleGAN, a new type of adversarial network for image generation that leverages two types of cycle-consistency. If we want to learn the conditional image distribution P(B | A), the imaget cycle consistency term encourages the network's encoder and generator to be able to yield B -> z -> B. where z are the GAN's latent noise units. The latent cycle consistency term encourages that z -> B ->z , i.e. the network should be able to generate an image conditioned on (z, A), then encode that image to recover z. Empirically, the authors show that the proposed setup yields diverse conditional image samples across several popular domains: line drawings -> shoe, line-drawings -> handbag, outdoor photographs night -> day, and building facade -> building photograph. I think this is a very nice result arising from an intuitive method. I am very curious how important it is for this method to have paired data across domains. Recently there have been many papers about learning conditional image distributions *without* domain alignments, using domain adversarial losses or cycle-consistency terms. I think the paper would be strengthened by carefully relating to this literature. Would it be possible to use the proposed method to learn *diverse* conditional image distributions without alignments?

Reviewer 3



Summary: The paper enforces a cycle consistency between the latent code and the image output of a generative model that is applied to image-image translation problems. The cycle consistency encourages invertibility of a generated output to the latent code, which addresses the popular issue of mode collapse in generative adversarial networks. The paper extends the pix2pix [1] framework of image-image translation by conditioning the generation of an image not only on an input image but also on a randomly sampled latent code. The low dimensional latent code captures the ambiguous aspects of the output image such as (depending on application) color, texture, etc.  While Isola et al. [1] observe that randomly sampling a latent code alone doesn’t help with mode collapse, the proposed approach applies a cycle consistency loss on both the input-output image pairs and the latent code - output pairs via alternating joint optimization. Strengths: * Mode collapse is a commonly occurring issue in generative adversarial networks, and a method to produce more diverse results could be potentially useful to a large community.  
 * The approach is a natural extension of the cycle consistency idea of [2] to address the issue of mode collapse. 
 * From the examples in the supplementary material it is clear that the proposed approach generates samples that are both realistic and diverse in the edges -> shoes task, when compared with baseline approaches such as pix2pix, pix2pix+noise, and an ablation of the overall approach, latent cycle GAN (described in 3.3).
 * The analyses in Section 4.2 are interesting. The analysis of the latent space reinforces the earlier claim regarding the poor distribution of latent codes when the model is trained using the Image Cycle. 
 * The paper is well written and easy to understand. 
 Clarification questions: * Are there any specific reasons or intuitions behind the decision to use the least squares loss in the experiment 3.1 (Equation (1)) instead of the discriminator loss used in [1]?
 * It is unclear whether the table in Figure 3., reports Google maps → Satellite images or the opposite transformation. 
 * Did the authors ensure that the performance of different methods in the table in Figure 3 are comparable with pix2pix? Specifically, are the ground truth images processed identically as [1]? Please note that [2] states that there are a few differences in preprocessing and hence the performance of methods in [2] are not comparable with [1]. 
 * Have the authors performed human-perception experiments on other datasets such as shoes? As a consequence of their semantic content, these datasets appear to contain more diverse images in terms of colors, textures etc., than aerial photos. 
 * It is unclear how the diversity score in the Table in Figure 3 is computed. Is it the cosine distance between vectors of neuron activations in a layer computed for pairs of images randomly sampled from the set B?
 Final recommendation: The paper proposes an approach to overcome the popular issue of mode collapse in generative adversarial networks which could be useful to further research into generating diverse samples in conditional generative models. References: [1] Isola et al.: Image-to-image translation with conditional adversarial networks. CVPR, 2017. [2] Zhu et al.: Unpaired image-to-image translation using cycle-consistent adversarial networks. arXiv preprint 2017.